# Interspecific two-dimensional visual discrimination of faces in horses (*Equus caballus*)

**Giulia Ragonese**[1], **Paolo Baragli**[2,3]*, **Chiara Mariti**[2], **Angelo Gazzano**[2], **Antonio Lanatà**[4], **Adriana Ferlazzo**[1], **Esterina Fazio**[1], **Cristina Cravana**[1]

1 Department of Veterinary Sciences, University of Messina, Messina, Italy, 2 Department of Veterinary Sciences, University of Pisa, Pisa, Italy, 3 Research Center "E. Piaggio", University of Pisa, Pisa, Italy, 4 Department of Information Engineering, University of Florence, Firenze, Italy

* paolo.baragli@unipi.it

**Data Availability Statement:** All relevant data are within the manuscript and its Supporting Information files.

## Abstract

In social animals, recognizing conspecifics and distinguishing them from other animal species is certainly important. We hypothesize, as demonstrated in other species of ungulates, that horses are able to discriminate between the faces of conspecifics and the faces of other domestic species (cattle, sheep, donkeys and pigs). Our hypothesis was tested by studying inter-and intra-specific visual discrimination abilities in horses through a two-way instrumental conditioning task (discrimination and reversal learning), using two-dimensional images of faces as discriminative stimuli and food as a positive reward. Our results indicate that 8 out of 10 horses were able to distinguish between two-dimensional images of the faces of horses and images showing the faces of other species. A similar performance was obtained in the reversal task. The horses' ability to learn by discrimination is therefore comparable to other ungulates. Horses also showed the ability to learn a reversal task. However, these results were obtained regardless of the images the tested horses were exposed to. We therefore conclude that horses can discriminate between two dimensional images of conspecifics and two dimensional images of different species, however in our study, they were not able to make further subcategories within each of the two categories. Despite the fact that two dimensional images of animals could be treated differently from two dimensional images of non-social stimuli, our results beg the question as to whether a two-dimensional image can replace the real animal in cognitive tests.

## Introduction

By discriminating between species, social animals can adapt their behavior according to the individual they meet [1]. Because of the great diversity in intraspecific phenotypes, species discrimination seems more similar to a categorization process than to a simple discrimination process [2]. Gregarious species with complex intra-specific communication patterns are likely to have the ability to categorize conspecifics [3], that is to place individuals belonging to the same species into the category "conspecifics" according to a number of common features. The

**Funding:** The authors received no specific funding for this work.

**Competing interests:** The authors have declared that no competing interests exist.

categorization of conspecifics can be achieved through multi-channel sensory modes [4]. However, animals might be able to recognize a conspecific using only one sensory modality [2].

The ability to visually discriminate conspecifics has been evaluated in many species, and the face seems to be a salient cue used to recognize individuals [5,6]. However, face recognition of conspecifics has been demonstrated only in a few species. A number of studies have been conducted on primates [7–9], and sheep (for a review see Tate et al. [10]). Face discrimination of conspecifics has also been demonstrated in cattle [1,2], dogs [11], birds [12–15], and invertebrates [16].

The recognition of faces and of many other types of three-dimensional objects seems to pose essentially the same problem: depending on the angle from which they are looked at or the position of their movable parts, both types of objects can project radically different images on the retina [17]. From a certain perspective, the recognition of common objects and faces seems to be supported by the same cognitive mechanisms [18].

Some researchers have focused on why facial recognition might be different from the recognition of common objects. In fact, it seems that the temporal cortex of mammals (humans, non-human primates and sheep) is equipped with specific neural circuits for facial recognition [19]. In addition, while most objects are slightly more difficult to recognize when they are shown upside down than when they are presented in the correct orientation [20], this inversion makes faces difficult to recognize, which reinforces the idea that faces are a "special" category [21,22].

Despite the growing number of studies on various animals, to our knowledge, no studies have evaluated the capacity of horses to use facial cues to discriminate conspecifics from other animals. Researchers have instead focused on olfactory recognition [23–25] and cross-modal recognition of conspecifics [4] and humans [26,27], and on human facial discrimination [28,29]. Evidence of facial discrimination would suggest that horses may use facial cues for the social recognition of individuals [30].

The study of discriminative learning in animals provides information on different areas of cognitive skills [31]. Discriminative learning tasks have been used to study color vision in horses [32–35] and their visual acuity [36]. In general, horses have the ability for discrimination and categorization learning [20,37–42]. They are also capable of solving reversal learning tasks, with some exceptions [43,44].

In the current study, we hypothesize that horses should be able to discriminate between two-dimensional images of faces of conspecifics and images of faces belonging to animals of four different species (donkeys, pigs, cows and sheep). This hypothesis is supported by outcomes from ungulate species, such as cattle [2] and sheep [5]. The aim of this study was therefore to evaluate the inter-specific visual discrimination capacities (inter-category discrimination) in horses through a two-way instrumental conditioning task (discrimination and reversal learning) using two-dimensional images of faces as discriminative stimuli and food as a positive reward. The use of facial cues from two dimensional social stimuli [30] should increase cognitive flexibility, enabling the horses to also perform the reversal task, when the stimuli provided have social valence.

In addition, to collect information on the categorization abilities in horses, we also analyzed the ability of the tested horses to categorize within two-dimensional images of conspecifics and two-dimensional images of different species (intra-category discrimination).

Intraspecific variation in behavior is ubiquitous among animals [45], and seems to be linked with cognitive style and personality [46,47], thus affecting the response to cognitive challenges at the individual level [48]. For instance, fast-exploring animals learn operant conditioning tasks more quickly than slow-exploring ones [49,50], whereas slow-explorers perform better in

reversal learning [51,52]. Animals that solve tasks quickly may also make more mistakes, while a greater amount of time may be required for more accurate decisions. This defines the speed-accuracy trade-off [53], which may help to understand an animal's individual cognitive style [54], possibly related to personality [48]. We therefore also analyzed the time required to perform the task. Our hypothesis is thus that a better performance in solving a task requires more time at an individual level.

## Materials and methods

### Ethics statements

This study was carried out in accordance with the recommendations of the Italian Animal Care Act (Decree Law 26/2014). The Ethical Committee on Animal Experimentation of the University of Messina approved the experimental protocol (Ref. No. 039/2020).

### Animals

For this study, 10 adult Franches-Montagnes horses (6 females and 4 males, aged 11.2±3.3 years) were used. The experiments took place at the "Azienda Agricola Traina" (Prizzi, Palermo, Italy). The horses were individually housed in boxes, and had paddock turnout for 6–8 hours a day, had the same feeding schedule, were in good health and were not affected by any stereotyped behavior. All the horses received the same training and handling, and were also managed by the same person in the every day management.

### Apparatus

The testing apparatus (Fig 1A) consisted of a rectangular wooden panel (cm 150x200) with two trapdoors (cm 80x50). Each trapdoor was lockable (closed or open) from behind [32]. Both trapdoors contained a transparent front cover where the stimulus picture was

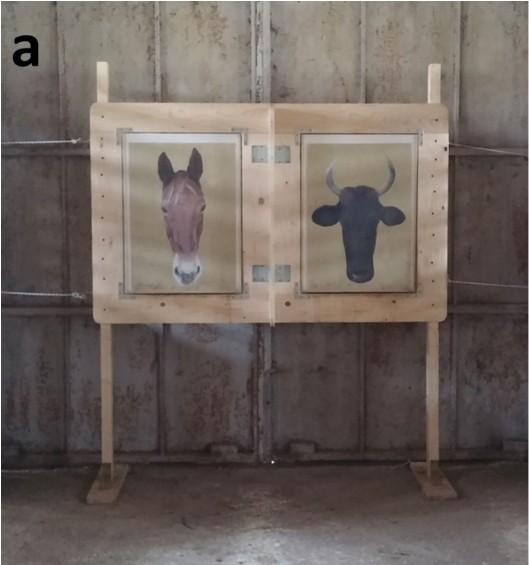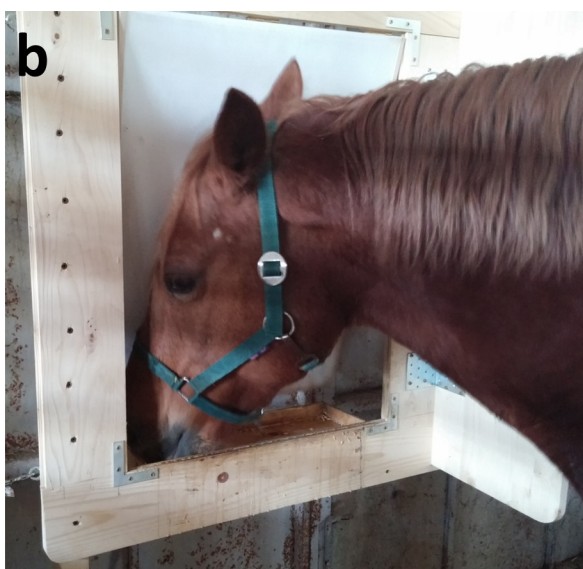

**Fig 1.** The testing apparatus (a) was similar to the one used by Macuda and Timney [32]. Two trapdoors were inserted into a rectangular wooden panel. Each trapdoor was lockable (closed or open) from behind. Both trapdoors contained a transparent front cover where the picture of the stimulus was inserted. The trapdoors together with the stimulus pictures, swung inward when the horse pushed them with its nose (b).

inserted. The trapdoors (including the stimulus pictures) swung inward when the horse pushed its nose against them (Fig 1B). This gave the horse access to a food reward placed on a shelf behind the trapdoor. The trapdoors were at the height of the horse's nose, and a wooden panel (height cm 150x40 cm depth) was placed between the trapdoors. This panel forced the horse to choose between one of the two stimulus panels at a minimum distance of 40 cm from the apparatus. The apparatus was located at the end of the stable corridor (14 m long by 3 m wide, Fig 2).

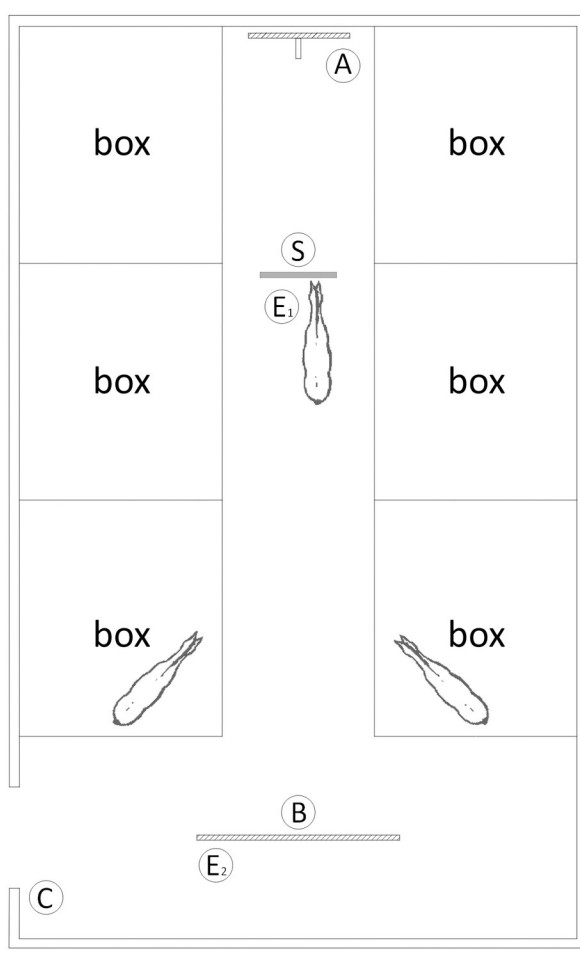

SCALE

1 m

A = TESTING APPARATUS
B = PARTITION
C = ENTRANCE TO TESTING AREA
S = STARTING POINT
E1 = EXPERIMENTER 1
E2 = EXPERIMENTER 2

Plan of the testing area, drawn to scale, with the horse in the starting position.

**Fig 2. The figure shows the apparatus, the starting position of the tested horse, and the positions of the two experimenters when the horse was released.**

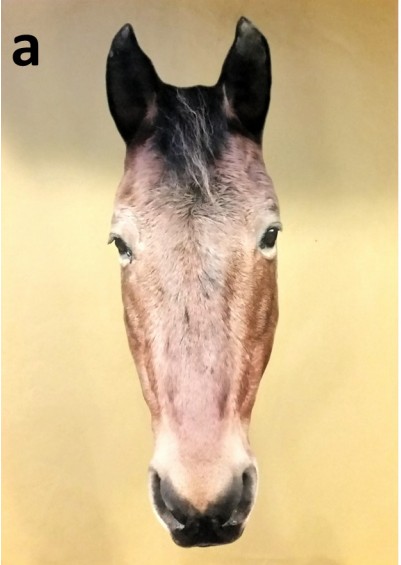
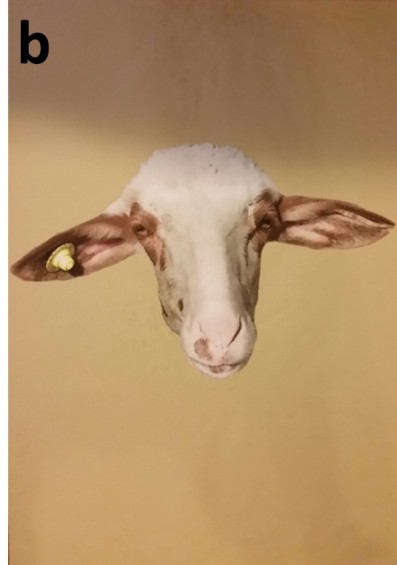

**Fig 3.** The digitized color pictures of a horse (a) and a sheep (b) used in the experimental design. The backgrounds of the original pictures were replaced by a uniform yellow background. Yellow was used for the picture background and it was chosen according to horses' color perception [35]. Yellow is also the color of common substrates in stables [1].

## Stimuli

The stimuli consisted in 20 digitized color pictures (70x50 cm). Ten pictures featured frontal views of the faces of different horses, while the other 10 picture were frontal views of faces of different domestic animals (2 cows, 3 sheep, 4 donkeys and 1 pig). In order to avoid biases induced by the pictures of familiar animals, all the faces of domestic animal species and horses were unknown to the subjects. The pictures were life-sized. See Fig 3 for an example and S1 Fig for the whole set of pictures.

## General procedure

The experiments were based on a two-way discrimination task involving the simultaneous discrimination of rewarded (S+) and non-rewarded (S-) stimuli. Responses were instrumentally conditioned to one stimulus during the training phase using food as the positive reward.

The tests were performed at the same time of day for each horse, in order to avoid changes in daily light. The horses were divided into five groups, two subjects each. Each group was tested consecutively on all experimental phases, and only when one group had completed all the phases, did the second one begin. Group order was randomly chosen.

Two female experimenters were involved: Experimenter 1 (E1, led the animal) and Experimenter 2 (E2, was responsible for programming the stimuli, filling the shelf with the right amount of food, indicating to E1 when to release the horse and recording the data on a paper sheet). At the beginning of each trial, the horse was led by E1 to the "starting point", 5 m from the apparatus. The starting point was indicated by a green band painted on the ground [32]. Following Hanggi [20], the horse was allowed to look at the stimuli for five seconds. Then E1 released the horse, enabling the animal to walk toward the apparatus. To avoid the "Clever Hans effect", E1 led the horse to the starting point and then stopped and turned round, facing the opposite direction. She also varied the side she was holding the horse throughout the trials. In addition, E1 wore a pair of earplugs, to avoid being distracted by external sounds. To

prevent any influence on the horse's decision, after having released the horse, E1 went behind a partition positioned on the other side of the corridor (Fig 2).

For each pair of stimuli, one stimulus (the correct choice) was consistently associated with a reward (S+) which consisted in 30 g of oat flakes. To prevent olfactory cueing, food was put behind both trapdoors. However, only the correct trapdoor provided access to food, while the other trapdoor (incorrect choice) was locked. By flipping a coin (heads = left; tails = right), the left–right position of the rewarded stimulus was semi-randomly balanced throughout the trials, with the constraint that the rewarded stimulus (S+) did not stay on the same side for more than two consecutive times [55]. After performing the trial, E1 led the horse behind a partition which served as a visual barrier during the placement of the next stimuli and food rewards. To reduce potential stress due to isolation, at the end of the corridor, on the opposite side to the experimental apparatus, two horses were left in the stable that were known to the horse being tested.

The experimental design consisted in a sequence of sessions. Each session consisted of 10 trials. Each trial consisted in a single choice made by the tested horse. Only the first choice of the horses was counted. For each subject, a block of three consecutive sessions (30 trials) was administered in a day (between 10:00 a.m. and 02:00 p.m.).

The whole experimental design included: 1) Familiarizing the horse with the apparatus;2) Check of the horse's discrimination ability, 3) Experiment 1 (training and generalization phases), 4) Experiment 2 (training and generalization phases of the reversal learning).

Apart from the familiarization procedure, in each phase (Check of the Discrimination ability, Training for Experiment 1, Generalization of the Experiment 1, Training for Experiment 2 and Generalization of the Experiment 2) the tested horse had to demonstrate their discrimination skills by reaching a criterion of success: eight correct choices (out of 10 trials) in two consecutive sessions. Horses were admitted to the next phase only when they had reached this score. If a horse failed to reach the score within 15 sessions, it did not continue to the subsequent phase. The experimental design was based on the studies by [1].

## Familiarization with the apparatus

Before starting the experiments, each subject went through a familiarization procedure [28]. On Day 1 both trapdoors were left open. E1 led the horse to the trapdoors five times and let the horse eat from both shelves for 5 s each. On Day 2, again for five times, both trapdoors were closed but left unlocked. E1 led the horse to an alternating trapdoor (sequence ABABA; A = Left, B = Right), opened it, and held it open in order for the horse to eat. On Days 3 and 4, E1 led the horse to the trapdoors 10 times; from the first to the eighth time, E1 held the horse at the starting point. After the 5-s observation period, E1 led the horse to the trapdoors (sequence ABBA-BAAB), opened them, and allowed a 5-s eating period while softly placing the trapdoor on the horse's muzzle. The last two times, after the 5-s observation period, E1 released the horse at the starting point, allowing the animal to choose between one of the two trapdoors. If the horse did not go to the trapdoors within 10 s, E1 led the animal to each trapdoor, opened it, and gently placed the trapdoor on the muzzle while the horse was eating. On Day 5, after the 5-s observation period, E1 released the horse at the starting point 10 times and allowed it to choose one of the two trapdoors. If the horse did not advance to the apparatus within 10 s, E1 turned the horse away from and waited for one minute before turning the horse around and starting again. No stimulus pictures were used during the familiarization phase.

## Discrimination ability check

After the familiarization, the horse was submitted to a simple two-choice discrimination task, in order to establish its basic ability to respond differentially to one of two stimuli. If the horse

was able to successfully perform this discrimination test, then errors committed during experiments could not be attributed to their inability to learn a two-choice discrimination task [56]. In this phase the stimuli consisted in pictures (70x50 cm) of two simple geometric shapes, an "X" and a circle "O". The stimulus associated with the food reward (S+) was the "X". The position of the two shapes was changed between the left and right, based on a semi-random criteria.

## Experiment 1

**Training phase of Experiment 1.**   A photograph of a horse (S+, picture number 1 in Supplementary File 1) and a photograph of a pig (S-, picture number 11 in Supplementary File 1) were used. The same pair of pictures was used throughout the training phase.

**Generalization phase of Experiment 1.**   All the stimuli were used, i.e. 10 pictures of the faces of different horses and 10 pictures of the faces of other domestic animals. Therefore, 18 new stimuli were introduced (in addition to the two stimuli already used in training phase). As in the training phase, the subjects were rewarded when they chose the picture of the horse's face. The pair of stimuli (one picture of a horse and one picture of another species) was extracted randomly and was changed at each trial.

## Experiment 2

**Training phase of Experiment 2.**   The procedure and the stimuli were the same as in the Training phase of Experiment 1. However, in the reversal learning, the picture of a sheep (S+, picture number 13 in Supplementary File 1) was rewarded instead of a horse (S-, picture number 5 in Supplementary File 1).

**Generalization phase of Experiment 2.**   The procedure was the same as in the generalization test in Experiment 1. The picture of the face of other species was the rewarded stimulus.

## Data collection and analysis

The number of sessions required to reach the criterion of success were recorded in the Training and Generalization phases of Experiments 1 and 2. We also recorded the total number of correct choices and the time required to make the choice in each trial in the Generalization phases of Experiments 1 and 2. Due to the non-normal distribution of data (Shapiro-Wilk test for non-normality, $p<0.05$) and the small sample size [57], the Wilcoxon Signed Ranks Test was performed to compare the number of sessions required for the criterion of success between the Training and Generalization phases in Experiments 1 and 2, and between the Generalization phases of Experiments 1 and 2. The performance at the individual level (number of trials with positive outcomes out of the total number of trials) in the Generalization phases of Experiments 1 and 2 was analyzed with the binomial probability test.

A Generalized Linear Mixed Model (GLMM) was computed in the Generalization phases of Experiments 1 and 2 in order to evaluate whether the choices (correct or incorrect) and the time required to make the choices were affected by the trial number, sessions, pictures of conspecifics, and pictures of different species.

Specifically, two GLMMs were implemented for each experiment. In the first GLMM, the dependent variable was the choice (correct or incorrect), while the predictors (fix factors) were the sessions, the pictures of the conspecifics and the pictures of the different species. The trials were taken into account as repeated measures. The subject identity was set as the random factor. A binary logistic regression model to predict the dependent variable was used. Specifically, a binomial distribution was used as the dependent variable distribution and the Logit as the link function [58].

In the second GLMM, the time required by the horse to make the choice was set as the dependent variable, while the fix and random effects were the same as in the first GLMM. A normal distribution for dependent variables was used with Identity as the link function [58]. In both models the Intercept was included in fix and random effects, while the variance component method for the random effect estimation was applied (for a review of the GLMM methodologies see [58]. Values of $p<0.05$ (two-tailed) were considered statistically significant. The analyses were performed with SPSS Statistics, version 26.0 (SPSS Inc., Chicago, Ill., USA).

## Results

Two horses out of 10 did not reach the success criterion (8 out of 10 correct answers in two consecutive sessions) during the Discrimination ability Check and were therefore excluded from the next step. Since these two subjects came from two different pairs of tested horses, the pairs of tested horses were randomly recreated. The eight remaining horses reached the criterion for success in the Discrimination ability Check and were admitted to the experimental phases. The acquisition curve at the individual level in the Generalization phases of both Experiments 1 and 2 is reported in Fig 4.

### Experiment 1

**Training phase of Experiment 1.** According to the criterion for success, the eight horses learned to discriminate between the horse's face and the pig's face.

**Generalization phase of Experiment 1.** When the horses were randomly exposed to pairs of pictures of 10 conspecifics and 10 other domestic animals, all subjects reached the criterion of success in fewer sessions than in the Training phase of Experiment 1 (Z = -2.379; p = 0.017; Fig 5A), and were therefore admitted to Experiment 2. In terms of positive outcomes over the

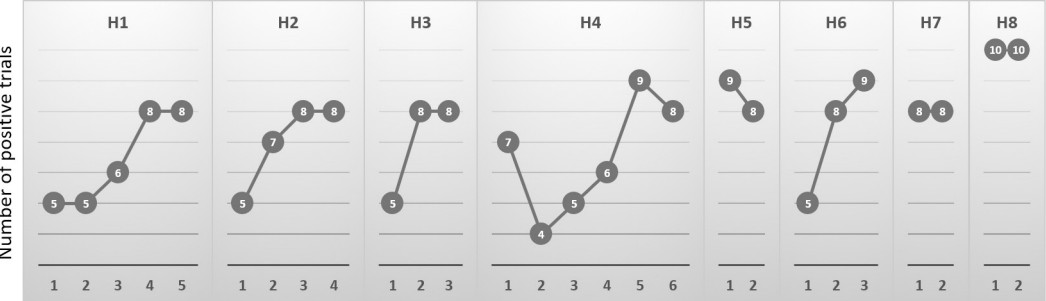

Sessions required to reach the criterion at individual level in the Generalization phase of the Experiment 1

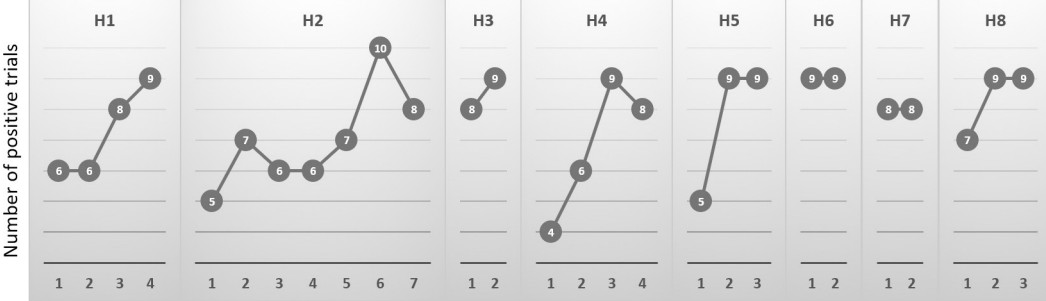

Sessions required to reach the criterion at individual level in the Generalization phase of the Experiment 2

**Fig 4. The acquisition curve of each horse is shown.** For each horse the number of positive trials in each session and the number of sessions required to reach the criterion are indicated for the Generalization phase of Experiments 1 and 2.

total number of trials performed, seven out of eight horses performed above the chance level (Table 1). The first GLMM (Akaike Information Criterion, AIC, for short data sample = 3552.604; Bayesian Information Criterion, BIC, for model comparison = 3590.035) revealed that none of the fixed or random factors affected the results (see S1 Appendix). With regard to the execution time in phase 3, there were large individual differences (3.3 up to10.0 s), and the second GLMM (AIC, corrected for short data sample = 707.796; BIC for model comparison = 745.226) revealed that the sessions affected the time required by the horse to make the choice (F = 5.127; df = 5–246; p<0.000; see S2 Appendix).

### Experiment 2

**Training phase of Experiment 2.** In the reversal learning, according to the success criterion, all subjects learned to discriminate between the pictures with the sheep's face and the horse's face.

**Generalization phase of Experiment 2.** When the horses were randomly exposed to pairs of pictures of 10 conspecifics and 10 other species in the reversal learning experiment, all subjects achieved the criterion in fewer sessions than in the Training phase of Experiment 2 (Z = -2.328; p = 0.020; Fig 5B) and all horses performed above chance level (Table 1). The first GLMM of the Generalization phase of Experiment 2 (AIC corrected for short data sample = 994.130; BIC for model comparison = 1031.511) revealed that the Sessions significantly affected the correctness of the choice (F = 2.517; df = 6–245; p = 0.022; see S3 Appendix). Also in the Generalization phase of Experiment 2 large individual variation was recorded with the execution time ranging from a minimum of 3.2 s to a maximum of 10.2 s. The second GLMM of the Generalization phase of Experiment 2 (AIC corrected for short data sample = 692.221; BIC for model comparison = 729.602) revealed that the Sessions affected the time required by the horse to make the choice (F = 6.887; df = 6–245; p<0.000; see S4 Appendix).

In both experiments, neither pictures of conspecifics nor the pictures of different species affected the results.

No statistical difference was reported when comparing the number of sessions required to reach the criterion of success between the Generalization phase of Experiment 1 and Experiment 2 (Z = -0.175; p = 0.861).

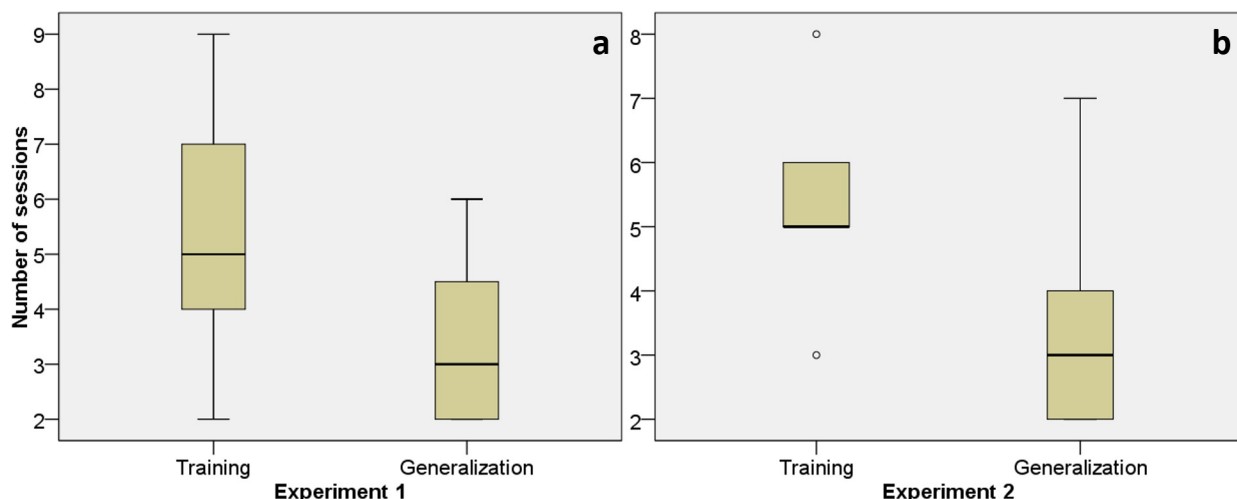

**Fig 5.** The number of sessions required by the tested horses to reach the success criterion in the Training and Generalization phases of Experiment 1 (Fig 5A) and Experiment 2 (Fig 5B).

**Table 1. The number of positive trials out of the total number of trials performed (between brackets) and the results of the binomial test are reported for each of the tested horses.**

| | Tested horses | | | | | | | |
|---|---|---|---|---|---|---|---|---|
| | H1 | H2 | H3 | H4 | H5 | H6 | H7 | H8 |
| Generalization phase of Experiment 1 | 32(50) | 28(40) | 21(30) | 39(60) | 17(20) | 22(30) | 16(20) | 20(20) |
| | z = 1.84 | z = 2.37 | z = 2.01 | z = 2.19 | z = 2.91 | z = 2.37 | z = 2.46 | z = 4.45 |
| | p = 0.065 | p = 0.017 | p = 0.043 | p = 0.027 | p = 0.003 | p = 0.016 | p = 0.011 | p = 0.000 |
| Generalization phase of Experiment 2 | 29(40) | 49(70) | 17(20) | 27(40) | 23(30) | 18(20) | 16(20) | 25(30) |
| | z = 2.69 | z = 3.23 | z = 2.91 | z = 2.06 | z = 2.74 | z = 3.35 | z = 2.46 | z = 3.47 |
| | p = 0.006 | p = 0.001 | p = 0.003 | p = 0.038 | p = 0.005 | p = 0.000 | p = 0.012 | p = 0.000 |

## Discussion

The results indicate that horses are able to discriminate between two-dimensional images of faces of conspecifics and images of faces belonging to other domestic species. After the training, during the generalization phases, all the horses were able to distinguish between the face of a horse from the face of another domestic animal, without having to learn the task again.

The number of sessions needed to reach the criterion was far more homogeneous in the generalization phases than the training phases (Fig 4). This suggests that the subjects did not necessarily require an equally high number of sessions to generalize what had been learned in the training phases, thus indicating a proper generalization ability, as already reported (see [31] for a review). This is strengthened by the GLMM of Generalization phase of Experiment 1, which indicates that the results were not influenced by the variables considered, including the repetition of the trials and sessions.

As part of a discriminative task, reversal learning is the ability of a subject to adjust their responses when the association between reward and stimulus is reversed. In our study, all the horses learned to discriminate between the stimuli, reaching the established criterion also in the reversal task, where the reward was associated with the faces of other animals. However, based on the GLMM in the Generalization phase of Experiment 2, the horses improved their performance as the sessions proceeded. This could indicate a learning effect and therefore that the reversal task is a more difficult test for horses than a simple discrimination task. This is in partial agreement with findings on reversal learning in *Equus caballus*. In fact, horses do not always perform correctly in a reversal task [43,59,60].

We found no significant difference between the Generalization phases of Experiments 1 and 2 in the number of sessions required to reach the criterion. This is in contrast with Coulon et al. [1], in which the cattle tested needed a lower number of sessions during the generalization phase of the discrimination task compared with the reversal task. Coulon et al. stated that this could be due to the fact that the formation of new stimulus-reward associations should be preceded by a deconditioning (the removal of the previous association).

The performance of the horses in our study could be explained by assuming that they were able to remove the previously formed association and, simultaneously, to form a new one faster than cattle. This is supported by Kendrick et al. [6], who found that the reversal test of the association between reward and stimuli had no effect on the discriminatory performance, when sheep were required to discriminate between familiar and unfamiliar faces. Furthermore, domestic horses are normally trained with the use of conditioned responses [61], therefore we cannot exclude that selective pressure of domestication could drive horses to learn faster conditioning and deconditioning responses.

When using two-dimensional stimuli, such as photographs, in assessing the ability of a species to visually recognize faces, it is important to ensure that the subjects are treating

photographs as representations of real animals and they are not just learning to distinguish between different visual patterns [2]. It is worth noting that during the first trial of the first session of the Generalization phase of Experiment 1, all subjects chose the picture of the horse. Furthermore, during the same trial, 5 out of 8 subjects sniffed around the whole picture of the horse's face and, in addition, one of the eight horses, during the fourth, fifth, sixth and seventh trials of the first session, showed an aggressive behavior (ears set back) towards the picture of the horse. It should also be emphasized that the horses did not show any difficulties in performing the task even when photographs of donkeys, which are morphologically similar to horses, were used.

These descriptive results, together with the statistical performance obtained, show that horses treated the faces belonging to their own species as equivalent, thus demonstrating a sort of mental representation of the conspecifics category. This is in line with a recent study in which horses demonstrated their ability to understand different two-dimensional facial expressions of conspecifics [30].

Several studies have shown that horses can distinguish between various photographs of three- and two-dimensional objects and recognize objects previously seen [20], and that they are also able to generalize the stimuli and to form categories [31,39,62–64]. However, Barbet (cited in [2], who studied the three-dimensional perception of two-dimensional images in the Guinea baboon (*Papio papio*), suggests that researchers should be prudent regarding the interpretation of the results of experiments in which two-dimensional images are used as stimuli in the place of real objects. In fact, the actual ability of animals to understand the relationship and differences between an object or a face and its two-dimensional representation is still unclear.

In both experiments in our study, neither the faces of conspecifics nor the faces belonging to other species affected the choice outcomes of the tested horses. This could be a demonstration of the discrimination and reversal abilities between the two categories (faces of conspecifics and faces of other animal species) in the tested horses. However, it seems that they were not able to categorize among the faces of different species and among the faces of conspecifics.

These results could be explained as follows: despite the fact that two dimensional faces (social stimuli) were treated differently to geometrical shapes, they do not fulfill the multisensorial information required in a social context and therefore cannot replace real animals.

Regarding the time required by the horse to make the choice in each trial, this increased as the sessions proceeded in both the Generalization phases of Experiments 1 and 2. However, this increase was not related to the subject's identity. Due to the criterion chosen, the last two sessions were always those with the highest performance (8 correct choices out of 10). This may indicate that the decision-making process requires some sort of "greater concentration" to become more accurate, which results in a longer time needed to make the choice. However, there was no evidence of a speed-accuracy trade-off at the individual level, as already reported in horses performing a spatial task [65].

## Conclusions

Our results indicate that horses are able to discriminate between two-dimensional images of conspecifics and those of other domestic animals, similarly to previous findings in other ungulates. In addition, despite some difficulties, in this study the horses tested also demonstrated their learning ability in the reversal task, in contrast to cattle.

A higher motivation of the horses toward social pictures rather than pictures of artificial shapes and colors could explain such conclusions. This supports the idea that, due to its social valence, the recognition of two-dimensional stimuli (faces) follows specific mind processes, which differ from the recognition of non-social stimuli [17,22]. Our horses however, did not

appear to be able to categorize among faces of conspecifics and among faces from different animal species. It is therefore probable that only the shape of the face drives their discrimination and reversal abilities. These conclusions highlight that specific investigations are required to understand whether two-dimensional social stimuli are treated differently from three-dimensional ones (real animal).

## Supporting information

**S1 Fig.**
(PDF)

**S1 Appendix.**
(PDF)

**S2 Appendix.**
(PDF)

**S3 Appendix.**
(PDF)

**S4 Appendix.**
(PDF)

**S1 Data.**
(XLSX)

## Author Contributions

**Conceptualization:** Giulia Ragonese, Paolo Baragli, Chiara Mariti, Adriana Ferlazzo, Esterina Fazio, Cristina Cravana.

**Data curation:** Giulia Ragonese, Paolo Baragli, Angelo Gazzano, Antonio Lanatà, Cristina Cravana.

**Formal analysis:** Angelo Gazzano, Antonio Lanatà, Cristina Cravana.

**Investigation:** Giulia Ragonese, Angelo Gazzano, Esterina Fazio, Cristina Cravana.

**Methodology:** Giulia Ragonese, Paolo Baragli, Chiara Mariti, Angelo Gazzano, Antonio Lanatà, Cristina Cravana.

**Project administration:** Adriana Ferlazzo, Esterina Fazio.

**Resources:** Antonio Lanatà, Adriana Ferlazzo, Esterina Fazio.

**Software:** Chiara Mariti, Antonio Lanatà.

**Supervision:** Paolo Baragli, Chiara Mariti, Angelo Gazzano, Adriana Ferlazzo, Esterina Fazio.

**Validation:** Chiara Mariti, Antonio Lanatà, Cristina Cravana.

**Writing – original draft:** Giulia Ragonese, Paolo Baragli, Chiara Mariti, Cristina Cravana.

**Writing – review & editing:** Paolo Baragli, Chiara Mariti, Angelo Gazzano, Adriana Ferlazzo, Esterina Fazio, Cristina Cravana.

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
