## [Decision Letter · Decision Letter 0]

15 Jul 2020

PONE-D-20-17626

Interspecific two-dimensional visual discrimination of faces in horses (Equus caballus)

PLOS ONE

Dear Dr. Baragli,

Thank you for submitting your manuscript to PLOS ONE. After careful consideration, we feel that it has merit but does not fully meet PLOS ONE’s publication criteria as it currently stands. Therefore, we invite you to submit a revised version of the manuscript that addresses the points raised during the review process.

I have been able to obtain expert on your study from 2 experts in animal behaviour. There are a number of low level cue confounds that have to be addressed. Thus if you require more time to enable revisions please request this. The manuscript if resubmitted will go back to both reviewers, and to proceed I expect very detailed responses to questions. 

We look forward to receiving your revised manuscript.

Kind regards,

Adrian G Dyer, Ph.D.

Academic Editor

PLOS ONE

Journal Requirements:

Reviewers' comments:

Reviewer's Responses to Questions

**Comments to the Author**

1. Is the manuscript technically sound, and do the data support the conclusions?

Reviewer #1: Partly

Reviewer #2: Partly

2. Has the statistical analysis been performed appropriately and rigorously? 

Reviewer #1: No

Reviewer #2: I Don't Know

3. Have the authors made all data underlying the findings in their manuscript fully available?

Reviewer #1: No

Reviewer #2: Yes

4. Is the manuscript presented in an intelligible fashion and written in standard English?

Reviewer #1: Yes

Reviewer #2: No

5. Review Comments to the Author

Reviewer #1: This manuscript demonstrates the capacity of horses to discriminate 2D faces pictures on the basis of the picture being a face of an unknown conspecific or from a different species. This study therefore both explore the capability of visual categorisation and of conspecific recognition based on facial visual cues. Both questions are of wide interest. However, as acknowledged by the authors themselves, it is not possible with the current state of data to make sure that the pictures were recognized as animal faces and not as any visual object. The task could indeed be solved by perceptual similarities within the horse faces category. Although some behaviours performed by the subjects in front of the conspecific faces and previous literature on object recognition suggest that horses may recognize the stimuli as representative of real animals, this point would need to be fully confirmed in further studies. Additionally, the horses were tested for their ability to reverse the learning task (chose now the heterospecific faces instead of the horse faces). The purpose of this reversal phase is not fully clear in the current version of the manuscript: was the aim to demonstrate the cognitive flexibility of horses independently of the species recognition question or was it a control for a natural attraction for horse faces ? In the latter case this was probably not the best method for such a control. It would have been much more appropriate as normally done in such categorisation task to randomize between subjects the reward category (conspecific or heterospecific). Such a procedure would have allowed to evidence or exclude any attractive bias for one category over the other without adding the noise of a reversal task in which subjects need to inhibit a learned attraction. The current set of data is consequently difficult to interpret as mixing 2 potential influences: conspecific attractiveness and reversal cognitive ability. An other crucial control necessary to conclude on categorisation ability is lacking: the within-category discrimination should be proved. If horses could not make the difference between the different face pictures then we are no longer in a categorisation task. I am aware that such an absence of discrimination is unlikely given the known visual ability of this species but this point should at least be raised convincingly in the manuscript. Similarly, in the simple discrimination task (‘X’ vs. ‘O’), not all subjects should have been rewarded for the ‘X’. A counter-balanced design should have been favoured to again avoid any potential preference bias to influence the results.

Other comments:

1. The results as reported (table 1) are not sufficient to evaluate the dynamic of the task acquisition. Indeed, applying a binomial test on all the choices is too primitive to account for an improvement in performance with time. I strongly suggest to provide the acquisition curves (% of correct choices over time, e.g. by session) and to apply a GLMM analysis on a binomial family with the choices (correct or incorrect) as the dependent variable, the trial number and potentially the species of the distractor stimuli as predictor and the subject identity as a random factor to account for the repeated measurement design. Such an analysis would allow to see if there was an improvement of performance as expected in case of learning and the potential effect of the distractor element. We could indeed hypothesize that the task was harder when the alternative face was from a donkey than from a pig for example, due to perceptual similarities. This information would contribute to a subtler interpretation of the results in term of a learned category over training or an innate pre-existing categorization. In the second case we could expect a rapid (maximum one session) acquisition of the task and no influence of visual similarity.

2. I appreciate that the matrix showing correlations between many variables was provided. However, this set of data should be introduced by the presentation of the hypothesis behind such correlations: what should we be looking at in particular? Which correlation should we expect in case of successful learning and so on… The question of a potential speed/accuracy trade-of should be addressed in particular. Finally, the results of this matrix should be further discussed.

3. I was surprised by the statement of a positive correlation between decision time and number of sessions. I do not see this result clearly in the matrix. Could you better justify your claim ?

4. Could you please provide the rationale behind the ET1 and ET2 phases? Why not starting with the full set of stimuli?

5. It seems that there are some confusions between the experimenters E1 and E2 in the text and on the figure. Please correct.

6. The acronyms: ET1, ET2, DT, EG1… are not easy to follow. Please consider replacing them by full words or more transparent acronyms.

Reviewer #2: The study conducted tested horses for their ability to discriminate the face of a horse from faces of other animals. I have some points below which need to be addressed before I can recommend this study for publication. Overall, the paper could do with revisions of language use and be reviewed for typos (point 3). I worry that size of the stimuli could have played a role in facial learning, for example the face of horses is much larger than that of pigs or sheep (Figure 3; E1T & E2T) and thus perhaps horses learnt size discrimination rather than facial discrimination? I also have some questions/concern about the methods which should be addressed.

1. Why was a yellow background used (Figure 3)? This should be explained in the text.

2. Figure 2 shows the position of E1 twice and misses the position of E2.

3. The paper could do with revisions on word use, sentence structure, and be re-read for typos. I have listed just a few examples here:

a. L133: "the horse was conducted by E1 to the “starting point”" - do you mean led?

b. Space missing at the end of line 244.

c. Double space in L252.

d. L251 - delete the word 'to'.

e. The wording of the abstract need to be reviewed. Remove the word 'also' from line 23. Line 29-30 is confusing - reword for clarity.

4. Why was a counter-balanced experimental design not used for the discrimination training? Horses were only trained to choose 'X' over 'O'. Could this have influenced visual preferences later?

5. Authors state that images were life-sized. Could this mean horses were choosing faces based on size? E.g. a horse's face is much large than the face of a sheep or pig.

6. The authors must include all other animal species shown to the horses.

7. More details need to be included in the methods for repeatability: i) were the horses able to get the reward from the correct stimulus if they chose incorrectly first, ii) was only the first choice of the horse counted if it made an error and then chose correctly in a single trial?

8. L214-216: Which took longer for horses to learn? DT or E1G and DT or E2G?

9. Can you elaborate on lines 219-220? I am unsure of what this sentence means. "Since these two subjects came from two different pairs of tested horses, the pairs of tested horses were randomly rebuilt."

10. Why was E2T performed with a sheep face but E1T performed with a pig face?

11. It is impossible for me to read Figure 5, so I cannot comment on this.

12. All stimuli used need to be shown in a figure in either the main manuscript or supplementary material.

13. Are the training/conditioning/experimental backgrounds of horses similar in this study?

6. PLOS authors have the option to publish the peer review history of their article (what does this mean?). If published, this will include your full peer review and any attached files.

Reviewer #1: No

Reviewer #2: No

---

## [Author Response · Author response to Decision Letter 0]

7 Dec 2020

Reviewer #1

1) This manuscript demonstrates the capacity of horses to discriminate 2D faces pictures on the basis of the picture being a face of an unknown conspecific or from a different species. This study therefore both explore the capability of visual categorisation and of conspecific recognition based on facial visual cues. Both questions are of wide interest. However, as acknowledged by the authors themselves, it is not possible with the current state of data to make sure that the pictures were recognized as animal faces and not as any visual object. The task could indeed be solved by perceptual similarities within the horse faces category. Although some behaviours performed by the subjects in front of the conspecific faces and previous literature on object recognition suggest that horses may recognize the stimuli as representative of real animals, this point would need to be fully confirmed in further studies.

R: Yes, we fully agree with this comment. At the present time, no final data on the topic are available. Furthermore, if horses are capable of high level cognitive abilities (such as cross modal recognition of conspecific and humans) they should also be able to discriminate between bi-dimensional pictures and real subjects. However, giving that a picture of a conspecific or of other animals could comprise some emotional involvement (as it happens in human) we cannot exclude that even bi-dimensional images of conspecifics and other animals are recognized by the horse and processed differently from geometrical shapes (as suggested in literature). However, further analysis (GLMM) have been performed in the revised ms to check for categorization abilities within categories (conspecifics and other species) and it seems not the case. Despite the fact that social pictures seems to be treated differently from bi-dimensional geometric shapes, we conclude that horses are able to discriminate between faces of conspecifics and faces of different species, but it seems that they were not able to categorize within each of the two groups of stimuli (see lines 357-364 and 378-385).

2) Additionally, the horses were tested for their ability to reverse the learning task (chose now the heterospecific faces instead of the horse faces). The purpose of this reversal phase is not fully clear in the current version of the manuscript: was the aim to demonstrate the cognitive flexibility of horses independently of the species recognition question or was it a control for a natural attraction for horse faces? In the latter case this was probably not the best method for such a control. It would have been much more appropriate as normally done in such categorisation task to randomize between subjects the reward category (conspecific or heterospecific). Such a procedure would have allowed to evidence or exclude any attractive bias for one category over the other without adding the noise of a reversal task in which subjects need to inhibit a learned attraction. The current set of data is consequently difficult to interpret as mixing 2 potential influences: conspecific attractiveness and reversal cognitive ability.

R: This comment is correct. From a conceptual point of view, probably both influences are true. To date, current bibliography indicates that horses are good performers with regards to discrimination tasks, while they perform poorly in reversal tasks. To our knowledge, these studies have been carried out using two-dimensional geometrical shapes. A higher performance in a reversal task where two-dimensional images of conspecific/heterospecific animals are used, could indirectly indicate that the recognition of two-dimensional images of social stimuli is not a simple visual discrimination process. However, the main purpose of the reversal learning we carried out was to understand the cognitive flexibility of horses, giving that other studies (where geometrical shapes were used) showed a certain impairment of horses’ ability to perform in this kind of cognitive tests. A sentence has been added to justify this approach in the introduction (see lines 79-81).

3) Another crucial control necessary to conclude on categorisation ability is lacking: the within-category discrimination should be proved. If horses could not make the difference between the different face pictures then we are no longer in a categorisation task. I am aware that such an absence of discrimination is unlikely given the known visual ability of this species but this point should at least be raised convincingly in the manuscript.

R: Yes, we agree with the reviewer’s comment. In the revised ms we added a GLMM statistics (as suggested) to check for categorization abilities within the categories “faces of conspecifics” and “faces of other species”. According to the results obtained, neither pictures of conspecifics nor pictures of other species affected the performance. We conclude that our horses were able to discriminate between conspecifics and other species but they were not able to categorize different conspecific subjects and different species (see lines 82-84; 357-361; 381-385).

4) Similarly, in the simple discrimination task (‘X’ vs. ‘O’), not all subjects should have been rewarded for the ‘X’. A counter-balanced design should have been favoured to again avoid any potential preference bias to influence the results.

R: The discrimination-training phase has been inserted as suggested in references only to verify that each one of the horses used was able to solve this kind of task. For this reason geometrical shapes have been chosen. We are sorry, it is not clear enough what kind of preference biases could exist between two-dimensional pictures of geometrical shapes and animals faces. A clear bias could be a side bias (if reinforcement was always administered on the same side). This bias has been prevented by alternating the left/right position of the rewarded shape. This information has been added to the revised ms (see lines 204-205).

Other comments:

5) 1. The results as reported (table 1) are not sufficient to evaluate the dynamic of the task acquisition. Indeed, applying a binomial test on all the choices is too primitive to account for an improvement in performance with time. I strongly suggest to provide the acquisition curves (% of correct choices over time, e.g. by session) and to apply a GLMM analysis on a binomial family with the choices (correct or incorrect) as the dependent variable, the trial number and potentially the species of the distractor stimuli as predictor and the subject identity as a random factor to account for the repeated measurement design. Such an analysis would allow to see if there was an improvement of performance as expected in case of learning and the potential effect of the distractor element.

R: We agree with the reviewer. These are good suggestions. The GLMM statistics and acquisition curve (Figure 4 in the revised ms, lines 253-254) were added to the revised ms (see lines 231-246; 270-276; 283-290). However, we invite the reviewer to reconsider her/his position about binomial test. Our suggestion is based on the fact that the outcome of such tests is essentially a yes/no one. Despite “primitive”, binomial test provide a strong and conservative analysis of the ability of horses to correctly perform in the task. For clarity purposes, only the performance of the Generalization phases of Experiments 1 and 2 are maintained in Table 1. We hope this is acceptable to the reviewer.

6) We could indeed hypothesize that the task was harder when the alternative face was from a donkey than from a pig for example, due to perceptual similarities. This information would contribute to a subtler interpretation of the results in term of a learned category over training or an innate pre-existing categorization. In the second case we could expect a rapid (maximum one session) acquisition of the task and no influence of visual similarity.

R: Yes, we fully agree. However, based on the GLMM analysis, none of the pictures belonging to different species affected the performance (not even those ones representing a donkey, which could have made the task harder to perform, as suggested). Therefore, no innate pre-existing categorization seem to be involved in the results obtained.

7) 2. I appreciate that the matrix showing correlations between many variables was provided. However, this set of data should be introduced by the presentation of the hypothesis behind such correlations: what should we be looking at in particular? Which correlation should we expect in case of successful learning and so on… The question of a potential speed/accuracy trade-of should be addressed in particular. Finally, the results of this matrix should be further discussed.

R: We fully followed the reviewer’s suggestion. For clarity purposes, the Kendal’s matrix has been removed. Possible speed/accuracy trade-off has been checked using a GLMM approach (see lines 241-246). Despite the fact that the time of the choice became higher as the sessions proceeded (suggesting a sort of “mental effort” to be even more accurate), the results of the GLMM do not reveal any influence of individuality on the time required to make the choice (see lines 373-376; 287-290).

8) 3. I was surprised by the statement of a positive correlation between decision time and number of sessions. I do not see this result clearly in the matrix. Could you better justify your claim?

R: Please, see the previous point, number 7. The Kendal’s matrix has been removed and replaced by a GLMM statistics. Essentially, a higher performance (i.e. an increase in correct choices along the sessions) is linked to an increase in the time required to make a choice, but this is not influenced by the subjects.

9) 4. Could you please provide the rationale behind the ET1 and ET2 phases? Why not starting with the full set of stimuli?

R: In discrimination tasks a training phase is generally required to teach tested animals to discriminate between two items of the checked categories. Only after passing the training phase, animals can perform a multi-item discrimination test. This is necessary when discrimination between multi-item categories is the goal of the study. This is a common procedures employed in several published papers studying discrimination abilities, including the paper by Coulon et al (2007) that was the paradigm we followed.

10) 5. It seems that there are some confusions between the experimenters E1 and E2 in the text and on the figure. Please correct.

R: The position of both experimenters indicated in the Figure 2 is their final position immediately before the horse was released. In the ms the effectiveness of the assignment of each experimenter has been provided in details. This could have contributed in create confusion. The assignments of both experimenters have been detailed further in the ms (see lines 150-160) and figure caption (see line 125).

11) 6. The acronyms: ET1, ET2, DT, EG1… are not easy to follow. Please consider replacing them by full words or more transparent acronyms.

R: Yes, we understand. Acronyms were removed and replaced by full words.

Reviewer #2

12) The study conducted tested horses for their ability to discriminate the face of a horse from faces of other animals. I have some points below which need to be addressed before I can recommend this study for publication. Overall, the paper could do with revisions of language use and be reviewed for typos (point 3).

R: Thanks for this suggestion. The ms has been reviewed by a native speaker with a professional background on scientific publication. A certificate has been submitted together with the other files.

13) I worry that size of the stimuli could have played a role in facial learning, for example the face of horses is much larger than that of pigs or sheep (Figure 3; E1T & E2T) and thus perhaps horses learnt size discrimination rather than facial discrimination? I also have some questions/concern about the methods which should be addressed.

R: Contrary to Coulon et al (2007), the images of the animals were life-sized. We thought this was necessary to align bi-dimensional images with what horses really see in their everyday life. What the reviewer says is probably correct. In fact, it is possible that the size is one of the criterion used to discriminate between different species. For this reason, we decided to maintain the real size. We hope the reviewer will accept this explanation. Furthermore, by performing GLMM we noted that the performance was not affected by the species represented in the pictures (neither conspecifics nor other species). Therefore, the size of the face is probably not involved in the discrimination process.

14) 1. Why was a yellow background used (Figure 3)? This should be explained in the text.

R: Yes, correct comment. A sentence has been added to Figure 2 caption to clarify why we used yellow as pictures’ background (see lines 137-139).

15) 2. Figure 2 shows the position of E1 twice and misses the position of E2.

R: We are sorry, this is not clear. In Figure 2 the position of both experimenters are reported at the moment in which the horse is released (E1 leading the horse, E2 behind the partition). We tried to explain this better in the ms (see lines 150-160) and in the caption of Figure 2 (see lines 125).

16) 3. The paper could do with revisions on word use, sentence structure, and be re-read for typos. I have listed just a few examples here:

17) a. L133: "the horse was conducted by E1 to the “starting point”" - do you mean led?

R: We made this correction (see line 153). Furthermore, the entire ms has been reviewed by a native speaker with a professional background on scientific publication. I hope this will help.

18) b. Space missing at the end of line 244.

R: It has been added.

19) c. Double space in L252.

R: The sentence has been removed in the revised ms.

20) d. L251 - delete the word 'to'.

R: Done.

21) e. The wording of the abstract need to be reviewed. Remove the word 'also' from line 23. Line 29-30 is confusing - reword for clarity.

R: Done, see line 21. Furthermore, based on the results obtained with the GLMM, the abstract has been reorganized (see lines 27-35).

22) 4. Why was a counter-balanced experimental design not used for the discrimination training? Horses were only trained to choose 'X' over 'O'. Could this have influenced visual preferences later?

R: Yes, in the ms we forgot to indicate that the left-right position of the two shapes was balanced across the trials according to a semi random criterion (see lines 204-205, see also point 4 of this reply to reviewers). However we would like to underline that the discrimination phase had the goal to just check whether all the horses involved were able to discriminate between two different stimuli. For this reason only one of the two shapes was reinforced (the same shape for all of the horses).

23) 5. Authors state that images were life-sized. Could this mean horses were choosing faces based on size? E.g. a horse's face is much large than the face of a sheep or pig.

R: See point 13. The images were life-sized to align the two-dimensional stimuli with what the horses see in their real life. This was our criterion for this choice. We hope this is acceptable.

24) 6. The authors must include all other animal species shown to the horses.

R: Yes, correct suggestion. The pictures of all the animals (including horses and other animal species) were added as supplementary material (supplementary S1 Figure).

25) 7. More details need to be included in the methods for repeatability: i) were the horses able to get the reward from the correct stimulus if they chose incorrectly first, ii) was only the first choice of the horse counted if it made an error and then chose correctly in a single trial?

R: Yes, this suggestion is correct. The horses were not able to get the reward when they made the incorrect choice. The incorrect-choice trapdoor was locked, therefore the horse could not have access to the reinforcement. Furthermore, the wooden panel positioned in the middle of the apparatus, between the two images, prevented the horse to eat the food in case of wrong choice (in Figure 1b you can see the partitioning wooden panel below the neck of the horse) and only the first choice was counted. All the requested information have been included in the ms (see lines 163-164 and 171).

26) 8. L214-216: Which took longer for horses to learn? DT or E1G and DT or E2G?

R: Due to individual variability it is not easy to reply to this question. The Discrimination Test (Check of the Discrimination ability in the revised ms, acronyms have been removed following other reviewers’ suggestions) was only implemented to verify the discrimination abilities in the tested horses (see also point 4 of this reply to reviewers). For this reasons and to avoid confusion, data collected in this phase have been removed from the ms.

27) 9. Can you elaborate on lines 219-220? I am unsure of what this sentence means. "Since these two subjects came from two different pairs of tested horses, the pairs of tested horses were randomly rebuilt."

R: To guarantee that the same horse was tested at the same day time, we divided the ten horses in 5 pairs. The pairs were tested in sequence. Only when the two horses had finished the session another pair was tested. Giving that two out of ten horses were excluded, the remained eight horses were remixed in four pairs.

28) 10. Why was E2T performed with a sheep face but E1T performed with a pig face?

R: The sheep and pig faces were randomly selected.

29) 11. It is impossible for me to read Figure 5, so I cannot comment on this.

R: In the revised ms the Kendal’s matrix has been replaced by more suitable statistic (GLMM). For details see lines 231-246; 270-276; 283-290.

30) 12. All stimuli used need to be shown in a figure in either the main manuscript or supplementary material.

R: Yes, done. See point 13 of this reply to reviewer and supplementary S1 Figure.

31) 13. Are the training/conditioning/experimental backgrounds of horses similar in this study?

R: The horses lived in the same stable, with common management regarding training, handling and background. They were also managed by the same person along the days. This information has been added to the ms (see lines 104-105).

---

## [Decision Letter · Decision Letter 1]

5 Feb 2021

Interspecific two-dimensional visual discrimination of faces in horses (Equus caballus)

PONE-D-20-17626R1

Dear Dr. Baragli,

We’re pleased to inform you that your manuscript has been judged scientifically suitable for publication and will be formally accepted for publication once it meets all outstanding technical requirements.

Kind regards,

Adrian G Dyer, Ph.D.

Academic Editor

PLOS ONE

Additional Editor Comments (optional):

Reviewers' comments:

Reviewer's Responses to Questions

**Comments to the Author**

1. If the authors have adequately addressed your comments raised in a previous round of review and you feel that this manuscript is now acceptable for publication, you may indicate that here to bypass the “Comments to the Author” section, enter your conflict of interest statement in the “Confidential to Editor” section, and submit your "Accept" recommendation.

Reviewer #1: All comments have been addressed

Reviewer #2: All comments have been addressed

2. Is the manuscript technically sound, and do the data support the conclusions?

Reviewer #1: (No Response)

Reviewer #2: (No Response)

3. Has the statistical analysis been performed appropriately and rigorously? 

Reviewer #1: (No Response)

Reviewer #2: (No Response)

4. Have the authors made all data underlying the findings in their manuscript fully available?

Reviewer #1: (No Response)

Reviewer #2: (No Response)

5. Is the manuscript presented in an intelligible fashion and written in standard English?

Reviewer #1: (No Response)

Reviewer #2: (No Response)

6. Review Comments to the Author

Reviewer #1: (No Response)

Reviewer #2: (No Response)

7. PLOS authors have the option to publish the peer review history of their article (what does this mean?). If published, this will include your full peer review and any attached files.

Reviewer #1: No

Reviewer #2: No

---

## [Editor Report · Acceptance letter]

9 Feb 2021

PONE-D-20-17626R1 

Interspecific two-dimensional visual discrimination of faces in horses (*Equus caballus*) 

Dear Dr. Baragli:

I'm pleased to inform you that your manuscript has been deemed suitable for publication in PLOS ONE. Congratulations! Your manuscript is now with our production department. 

Kind regards, 

on behalf of

Dr. Adrian G Dyer 

Academic Editor

PLOS ONE